# The Use of a Fractional Factorial Design to Determine the Factors That Impact 1,3-Propanediol Production from Glycerol by *Halanaerobium*
*hydrogeniformans*

**DOI:** 10.3390/life6030035

**Published:** 2016-08-20

**Authors:** Shivani Kalia, Jordan Trager, Oliver C. Sitton, Melanie R. Mormile

**Affiliations:** 1Department of Biological Sciences, Missouri University of Science & Technology, Rolla, MO 65401, USA; sk73f@mst.edu (S.K.); jat6y5@mst.edu (J.T.); 2Department of Chemical and Biochemical Engineering, Missouri University of Science & Technology, Rolla, MO 65401, USA; ocs@mst.edu

**Keywords:** glycerol, 1,3-propanediol, haloalkaliphilic bacteria, fractional factorial design

## Abstract

In recent years, biodiesel, a substitute for fossil fuels, has led to the excessive production of crude glycerol. The resulting crude glycerol can possess a high concentration of salts and an alkaline pH. Moreover, current crude glycerol purification methods are expensive, rendering this former commodity a waste product. However, *Halanaerobium hydrogeniformans*, a haloalkaliphilic bacterium, possesses the metabolic capability to convert glycerol into 1,3-propanediol, a valuable commodity compound, without the need for salt dilution or adjusting pH when grown on this waste. Experiments were performed with different combinations of 24 medium components to determine their impact on the production of 1,3-propanediol by using a fractional factorial design. Tested medium components were selected based on data from the organism’s genome. Analysis of HPLC data revealed enhanced production of 1,3-propanediol with additional glycerol, pH, vitamin B_12_, ammonium ions, sodium sulfide, cysteine, iron, and cobalt. However, other selected components; nitrate ions, phosphate ions, sulfate ions, sodium:potassium ratio, chloride, calcium, magnesium, silicon, manganese, zinc, borate, nickel, molybdenum, tungstate, copper and aluminum, did not enhance 1,3-propanediol production. The use of a fractional factorial design enabled the quick and efficient assessment of the impact of 24 different medium components on 1,3-propanediol production from glycerol from a haloalkaliphilic bacterium.

## 1. Introduction

Humanity’s increasing consumption of energy requires the expenditure of resources. However, primary resources, i.e., fossil fuels, are finite and limited. To address this problem, the scientific community is working to develop effective methods for generating renewable sources of energy. Biodiesel is one of these emerging alternatives. However, a fundamental problem with biodiesel is that it leads to large amounts of waste products as almost 10% (*w*/*w*) glycerol is produced as the main byproduct [1]. Pure glycerol has numerous applications in food, pharmaceutical and cosmetic industries. Moreover, it can be converted to various chemical intermediates such as 1,3-propanediol (PDO), poly-hydroxyalkanoates, epichlorohydrin, acrylic acid, polyhydroxybutyrate etc. [2]. On the other hand, glycerol obtained from biodiesel is in crude form and has many impurities. Hansen et al. [3] performed a study to determine the composition of crude glycerol samples obtained from the different biodiesel plants. They demonstrated that glycerol content in the samples ranged from 38% to 96%, with some samples comprising up to 14% methanol and 29% ash. Furthermore, crude glycerol has high pH value of 10, and salt concentration of 3%. Therefore, purification and neutralization of crude glycerol, prior to its industrial utilization, is essential and leads to increased costs of biodiesel production [4].

Effective ways to use crude glycerol need to be developed to reduce the cost of biodiesel production, thus enhancing the growth of biodiesel industries. One such process is the conversion of crude glycerol to a chemical commodity, PDO. 1,3-propanediol has various industry applications, as it is used as a chemical intermediate for the synthesis of cosmetics, lubricants, adhesives, perfumes and laminates [5]. It is also used to make the polymer polytrimethylene terephthalate, a commodity utilized by the carpet and textile industries [6]. Previous studies have revealed the role of microorganisms such as *Klebsiella*
*pneumoniae* [7] and *Clostridium*
*butyricum* [8] can have in the production of PDO. *Citrobacter*
*freundii*, strain FMcc-B 294 (VK-19), is also capable of converting biodiesel-derived glycerol to PDO at an initial pH of 6.2. Additionally, this organism can produce PDO with no unexpected metabolic products when cultivated under non-sterile conditions [9]. Kivistö et al. [10] found that vitamin B_12_ was responsible for diverting electrons from H_2_ production to PDO production by *Halanaerobium*
*saccharolyticum* subsp. *saccharolyticum*. However, neutralization of the crude glycerol has to be done before it can be metabolized by these organisms.

Our study was performed with a haloalkaliphilc bacterium, *Halanaerobium hydrogeniformans*, isolated from Soap Lake in Washington State [11]. *H. hydrogeniformans* possesses the ability to convert crude glycerol into PDO. It should be noted that optimum growth conditions for this bacterium are: pH of 11, NaCl concentration of 7.5%, and a temperature of 30 °C [12]. Moreover, this bacterium is capable of growing at a broader pH range (7.5–12) and salt range (2.5%–15%) in comparison to other PDO-producing bacterial species. This property of *H. hydrogeniformans* makes it ideal for the conversion of crude glycerol into PDO. The use of this organism removes the need for the neutralization of crude glycerol, possibly making PDO production a cost-effective approach.

Genome data analysis of *H. hydrogeniformans* revealed that it possesses the genes for PDO production [11]. In general, glycerol to PDO production pathway entails two steps (Figure 1). In the first step, a water molecule is removed from glycerol, forming 3-hydroxypropionaldehyde, catalyzed by the enzyme glycerol dehydratase. In the following step, enzyme 1,3-propanediol dehydrogenase regenerates NAD^+^ needed by cell through the oxidation of NADH_2_ and produces PDO. Glycerol dehydratase is a vitamin B_12_ dependent enzyme. Moreover, *H. hydrogeniformans* possesses vitamin B_12_ synthesis pathway. However, earlier experiments performed with *H. hydrogeniformans*, on glycerol fermentation, suggest that the production of PDO almost doubles when media is supplemented with vitamin B_12_. This previous work also indicated that *H. hydrogeniformans* had a 60.3% conversion of glycerol in our standard growth media [14].

Therefore, the primary objective of this study was to determine the medium components that could have significant impact on the production of PDO from glycerol by *H. hydrogeniformans*. Furthermore, potential sources of nitrogen and sulfur in the media were evaluated. On the basis of standard medium composition, a total of 24 factors were evaluated for their effect on PDO production. The experimental design for these factors was generated by using a statistical approach, fractional factorial design. This study suggests that eight factors: glycerol, vitamin B_12_, pH, ammonium ion, sodium sulfide, cysteine, iron and cobalt have significant impact on the production of PDO.

## 2. Materials and Methods

### 2.1. Growth Medium Preparation

Cultures of *Halanaerobium hydrogeniformans* were maintained in the laboratory by using a standard medium composition i.e., 70 g sodium chloride (NaCl), 40 g sodium carbonate (Na_2_CO_3_), 6.3 g potassium phosphate dibasic (K_2_HPO_4_), 1 g yeast extract, glycerol 5 *w*/*v*% std (543 mM), 10 mL basal medium stock solution and 10 mL trace mineral solution in a 1 L solution [12]. The composition of 1 L trace mineral solution was: 30 mg magnesium sulfate heptahydrate (MgSO_4_·7H_2_O), 23.6 mg disodium ethylene diaminetetraacetatedihydrate (EDTA), 10 mg sodium chloride (NaCl), 6.4 mg manganous chloride (MnCl_2_·4H_2_O), 1.3 mg zinc chloride (ZnCl_2_), 1 mg ferrous sulfate heptahydrate (FeSO_4_·7H_2_O), 1 mg calcium chloride dihydrate (CaCl_2_·2H_2_O), 1 mg cobalt chloride hexahydrate (CoCl_2_·6H_2_O), 0.3 mg nickel sulfate hexahydrate (NiSO_4_·6H_2_O), 0.25 mg sodium molybdate dihydrate (Na_2_MoO_4_·2H_2_O), 0.25 mg sodium tungstate dihydrate (Na_2_WO_4_·2H_2_O), 0.1 mg aluminium potassium sulfate hydrate (AlK(SO_4_)_2_·12H_2_O), 0.1 mg boric acid (H_3_BO_3_) and 0.07 mg cupric chloride dihydrate (CuCl_2_·2H_2_O). The 1 L of basal medium stock solution contained 50 mg ammonium nitrate (NH_4_NO_3_), 8.5 mg magnesium chloride hexahydrate (MgCl_2_·6H_2_O), 7.5 mg silicon oxide (SiO_2_), 4.5 mg manganous sulfate monohydrate (MnSO_4_·H_2_O), 4.2 mg calcium chloride dihydrate (CaCl_2_·2H_2_O), 4 mg methylene blue (as an oxygen indicator) and 1.8 mg ferrous sulfate heptahydrate (FeSO_4_·7H_2_O). The resulting medium was made anaerobic by boiling and degassing under a stream of N_2_. As the temperature of the media dropped, 0–100 µg vitamin B_12_, 0.025 g Na_2_S (as a reductant), 0.025 g cysteine (as a reductant) per liter were added. The flask containing cooled medium was transferred into an anaerobic chamber under a forming gas (N_2_ (95%):H_2_ (5%)) atmosphere. In the anaerobic chamber, 50 mL amounts of the medium were dispensed into 160 mL serum bottles, sealed with butyl rubber stoppers held in place with aluminum crimp seals. The medium was sterilized by autoclaving at 121 °C/15 PSI for 20 min. The headspace gas in the serum bottles was exchanged aseptically with 100% nitrogen and vitamin B_12_, 0–100 µg, was added. The medium was inoculated with 10% inoculum from previous stock cultures. Cultures were incubated at 30 °C in a shaking incubator.

### 2.2. Experimental Design to Determine Impact of Medium Composition on PDO Production

On the basis of standard medium composition, a total of 24 (23 + Na:K) factors were considered to evaluate their impact on PDO production. Table 1 lists the experimental variables or factors, each identified by using a capital letter. For screening purposes, two levels of each factor, either high or low, were used to evaluate its significance in PDO production, as shown in the Table 1. These high and low concentrations were chosen to be sufficiently different to ensure a measurable change in response would occur if the factor was important for the production of PDO.

### 2.3. Fractional Factorial Design of Experiments to Determine Impact of Medium Composition on PDO Production

A screening experiment was implemented to determine the effect of each factor and the interactions among them by using a fractional factorial design approach. In this design, there are 2^k−p^ experiments where p defines the 1/2^p^ fraction of the total number of experiments. A set of 32 experiments was used to determine the effect of each factor. This design does not consider the effect of interfactor interactions on the PDO production. For *H. hydrogeniformans*, a significant interaction among the factors are not expected, as the bacteria either requires a factor or does not. Therefore, a classical resolution III design, also labeled as $2_{III}^{24-19}$ design, was used. The interaction among factors was considered negligible.

These 32 experiments were performed in triplicates and were divided into four blocks. Each block included a total 24 tests. Each of the 24 factors were taken in two concentrations either high or low level.

### 2.4. Optimization Tests Procedure

Test solutions were prepared to achieve the concentration of each tested component as prescribed in Table 1. The pH of these solutions were adjusted by using two different base stock solutions. One stock solution had a Na:K ratio of 12.5 and the other had a Na:K ratio of 50 to match the Na:K ratios to be tested. Test medium was made anaerobic by boiling and sparging with argon gas. Anaerobic medium was transferred to the anaerobic chamber and 50 mL aliquots of medium were transferred to 160 mL serum bottles. In each serum bottle, cysteine and sodium sulfide solutions were added. Serum bottles containing test medium were then sealed with butyl rubber stoppers held in place with aluminum crimp seals and autoclaved. After cooling, the serum bottles were inoculated with 1 mL of inoculum and placed onto a shaking incubator at 30 °C for three days. After three days, serum bottles were removed and samples were analyzed for PDO production by using HPLC.

For HPLC analysis, samples were filter sterilized by using 0.45 µM PTFE filters. Filtered samples were injected onto an Aminex HPX-87H, 300 × 7.8 mm column (BioRad, Hercules, CA, USA). The column’s temperature was maintained at 50 °C. Here, 2.5 mM·H_2_SO_4_ was used as mobile phase and its flow rate was 0.6 mL/min at 2.2 MPa pressure. For the sample analysis, two types of detectors; a UV 231 (at 210 nm) and refractive index monitor were used.

## 3. Results

The statistical experimental design resulted into 32 different screening tests, run in blocks (Table 2). Each block contained eight different test conditions. Experiments were run in triplicates. For each screening test, the media were inoculated with *H. hydrogeniformans* and incubated for 3 days. The production of PDO in each culture was determined by HPLC analysis.

### 3.1. Screening Tests and HPLC

The amount of PDO produced during each test is reported in Table 2. Yield was calculated as the moles of PDO produced per the moles of the initial glycerol concentration. Maximum PDO yield was observed when all the factors were added in their high concentration. The lowest yield was observed when low levels of glycerol, vitamin B_12_, nitrate, sulfate, sodium sulfide, cysteine, magnesium, silicon, iron, cobalt, tungstate, and copper were provided. This indicates that these components are essential during the production of PDO by *H. hydrogeniformans*.

### 3.2. Effect Tests

Based on genome data analysis, it was predicted that ammonium ion, nitrate ion, sulfate ion, Na:K ratio, zinc, iron, nickel, and tungstate might positively impact PDO production [11]. To confirm these predictions, an experimental design was generated. This experimental design was created by using fractional factorial design approach, in which individual effect of each variable or factor was determined. HPLC data for the PDO yield from the screening tests was fed into JMP Pro 12.1 software (http://www.jmp.com/en_us/software/jmp-pro.html) for the multiple linear regression analysis. This model generates data for the effect test, on the basis of PDO yield obtained. This analysis works on the null hypothesis (*p*-value > 0.05), that each factor has no significance and the alternate hypothesis (*p*-value < 0.05), where each factor is significant. Here, results from the effect tests are presented in the Table 3 that lists all the 24 factors and their respective *p*-value. The *p*-value for glycerol, vitamin B_12_, pH, ammonium ion, sodium sulfide, cysteine, iron, and cobalt were less than 0.05 for each. Therefore, the null hypothesis is rejected for each and the alternate hypothesis is accepted, indicating that these factors have significant impact on the production of PDO at the 95% confidence level. In contrast, *p*-value for nitrate ion, sulfate ion, phosphate ion, Na:K ratio, chloride, calcium, magnesium, silicon, manganese, zinc, borate, nickel, molybdenum, tungstate, copper, and aluminum was more than 0.05. Therefore, the null hypothesis is accepted, and these factors did not have a significant impact on the production of PDO at the 95% confidence level.

There is another factor provided in Table 3, the Block factor. As previously mentioned, screening tests were divided into four blocks and each block included eight experiments. These experiments were performed separately. The block factor is used to determine if any difference in results was due to performing these screening test on different days. The *p*-value for the block factor is 0.6052, i.e., more than 0.05. Thus, the block factor had no significant impact and there was consistency in test measurements.

The results from the fractional factorial design of these experiments also indicated that two factor interactions occurred for some of the main factors. This interaction tested whether a high or a low value of a factor had any impact on the other factor. This design works on the assumption that all two factor interactions are not important. Previous studies have shown the significance of factors such as glycerol, vitamin B_12_ and pH on the production of PDO form glycerol by *H*. *hydrogeniformans*. The interaction between glycerol * pH, vitamin B_12_ * pH, pH * MoO_4_ and NH_4_ * MoO_4_ were identified. *p*-value for the each of these interaction was more than 0.05, indicating that these interactions are not significant.

## 4. Discussion

### Summary of the Optimized Media Conditions

Microbial mediated conversion of crude glycerol into PDO is an effective way to utilize waste glycerol. Previous experiments performed in our lab showed significant production of PDO from glycerol by *H. hydrogeniformans* [14]. The primary objective of this study was to determine the impact of 24 different factors on the production of PDO. These factors were selected on the basis of standard medium components for the growth of *H. hydrogeniformans* [12]. From the genome information of this bacterium, factors such as ammonium ion, nitrate ion, sulfate ion, Na:K ratio, zinc, iron, nickel and tungstate were anticipated to be important for the PDO production [11]. It was anticipated that ammonium ion and nitrate ion are preferred sources of nitrogen, sulfate ion is the source of sulfur, and Na:K ratio is essential to maintain the ability for this organism to take up potassium for maintenance of its osmotic balance. The trace elements zinc, iron, nickel, and tungstate were thought to be crucial for the enzymatic activity.

Through statistical analysis, 8 of 24 factors were found to significantly enhance PDO production. These factors were glycerol, vitamin B_12_, pH, ammonium ion, sodium sulfide, cysteine, iron, and cobalt. Glycerol is the initial substrate in this pathway, thereby, higher concentration of glycerol would lead to higher production of end-product PDO. In this pathway, glycerol dehydratase converts glycerol into an unstable intermediate 3-hydroxypropionaldehyde (Figure 1). This intermediate is the precursor for PDO and the reaction is catalyzed by PDO dehydrogenase enzyme. Glycerol dehydratase is a vitamin B_12_ dependent enzyme, indicating that vitamin B_12_ a significant factor. Previous experiments performed with vitamin B_12_ demonstrated that *H. hydrogeniformans* can produce PDO even in the absence of B_12_. It appears that this bacterium possesses vitamin B_12_ synthesis pathway as evidenced in its genome. However, the supplementation of vitamin B_12_ doubles the concentration of PDO produced [14].

In the standard growth medium, ammonium ion and nitrate ion are supplemented as the potential nitrogen sources. In order to identify the nitrogen source utilized by bacteria, ammonium ion and nitrate ion were kept as separate factors. Generally, microorganisms prefer ammonium over nitrate as the nitrogen source [15]. The exact mechanism involved in the effect of ammonium ion on the production of PDO is still unknown. However, the evidence for the impact of ammonia limited media on the glycerol dissimilation to produce PDO has also been provided by Zheng et al. [16]. Furthermore, *H. hydrogeniformans* is an anaerobic bacteria and requires a reducing environment. In the standard growth medium of this bacterium, sodium sulfide and cysteine were added as the reducing agents that reduce the oxidation-reduction potential of the media. This makes these components essential for the growth of the bacterium and thereby, the production of PDO. Additionally, *H. hydrogeniformans* grows preferentially at a pH of 11 and produces more PDO under highly alkaline conditions.

Microbial glycerol metabolism under anaerobic conditions occurs mainly via enzyme-mediated oxidative-reductive pathways [17]. In general, the activity of enzymes is attributed to the availability of cofactors or metal ions. Our results revealed the significance of two cofactors, iron and cobalt, in the production of PDO. As mentioned before, glycerol dehydratase is a vitamin B_12_ dependent enzyme and typically cobalt is an integral part of vitamin B_12_’s structure. Therefore, concentration of cobalt in the media affects the structure and synthesis of vitamin B_12_, and subsequently, the production of PDO. Moreover, iron is an essential cofactor for the activity of many alcohol dehydrogenases. One such iron dependent alcohol dehydrogenase has been observed in the *C. acetobutylicum* [18]. In conclusion, selected medium components in higher concentrations significantly enhance the production of PDO by *H. hydrogeniformans*.

## Figures and Tables

**Figure 1 life-06-00035-f001:**
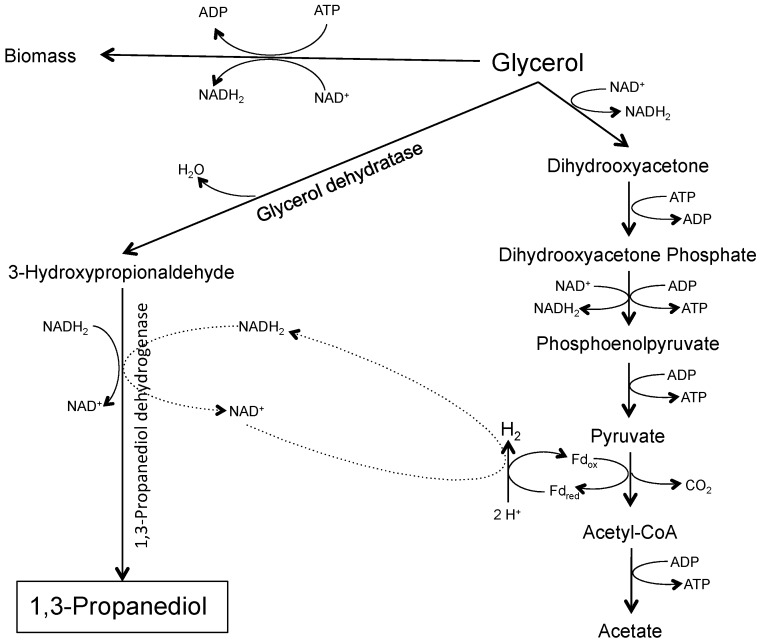
1,3-propanediol production pathway. Modified from Zeng [13].

**Table 1 life-06-00035-t001:** Factors for screening experiments.

Number	Factor	Name	Variable	Std. Media	Low Value (−1)	High Value (+1)
1	A	GLYC	Glycerol	542,947 µM	500,000 µM	2,000,000 µM
2	B	B_12_	Vitamin B_12_	100 µg/L	0 µM, 0.0M	500 µg/L, 0.3689 µM
3	C	pH	pH	10.85	9.0	10.85
4	D	NH_4_	Ammonium ion	624 µM	0 µM	3000 µM
5	E	NO_3_	Nitrate ion	624 µM	0 µM	3000 µM
6	F	SO_4_	Sulfate ion	160 µM	160 µM	800 µM
7	G	PO_4_	Phosphate ion	36,169 µM	40,000 µM	200,000 µM
8	H	NaK	Sodium:potassium ratio	27	12.5	50.0
9	J	NaS	Sodium sulfide	104 µM	50 µM	200 µM
10	K	Cys	Cysteine	142 µM	75 µM	300 µM
11	L	Cl	Chloride	1,198,371 µM	600,000 µM	2,400,000 µM
12	M	Ca	Calcium	35 µM	0 µM	500 µM
13	N	Mg	Magnesium	164 µM	0 µM	1000 µM
14	O	Si	Silicon	125 µM	0 µM	500 µM
15	P	Mn	Manganese	59 µM	0 µM	500 µM
16	Q	Zn	Zinc	9.5 µM	0 µM	100 µM
17	R	Fe	Iron	10.0 µM	0 µM	100 µM
18	S	Co	Cobalt	4.2 µM	0 µM	100 µM
19	T	BO_3_	Borate	1.6 µM	0 µM	10 µM
20	U	Ni	Nickel	1.1 µM	0 µM	10 µM
21	V	MoO_4_	Molybdenum	1.0 µM	0 µM	10 µM
22	W	WO_4_	Tungstate	0.8 µM	0 µM	10 µM
23	X	Cu	Copper	0.4 µM	0 µM	5 µM
24	Y	Al	Aluminum	0.2 µM	0 µM	2 µM

**Table 2 life-06-00035-t002:** Serum bottle and experimental block set-up with the average amount and standard deviation for three replicates for each of the tests run.

Test	Pattern	Block	Average PDO Yield (Percentage of Initial Glycerol Converted to PDO)	Std. Dev.
1,37,71	−−++−−++−−++−−++−−+++−−+	1,5,9	15.36	±0.62
2,40,68	−−−++−++−+−−+−++−+−−−++−	1,5,9	17.64	±0.78
3,34,66	+++−+−−−−−−−−+++++++−−−−	1,5,9	19.73	±1.27
4,36,72	+++−−++++++++−−−−−−−−−−−	1,5,9	19.90	±1.27
5,33,67	++−−−−−−−++++++++−−−++++	1,5,9	18.87	±0.65
6,39,65	++−−+++++−−−−−−−−+++++++	1,5,9	20.10	±0.23
7,38,69	−−−+−+−−+−++−+−−+−++−++−	1,5,9	17.80	±0.67
8,35,70	−−++++−−++−−++−−++−−+−−+	1,5,9	20.34	±0.93
9,46,76	+−+−+++−−++−−−−++−−+++−−	2,6,10	17.72	±0.83
10,43,79	−+−+++−+−−+−++−+−−+−+−+−	2,6,10	16.06	±0.97
11,42,75	+−−−+−−++++−−++−−−−+−−++	2,6,10	16.82	±0.17
12,41,80	−+−+−−+−++−+−−+−++−++−+−	2,6,10	22.42	±0.89
13,47,78	+−+−−−−++−−++++−−++−++−−	2,6,10	18.12	±0.65
14,45,73	−++++−+−+−+−+−+−+−+−−+−+	2,6,10	19.64	±0.51
15,44,77	+−−−−++−−−−++−−++++−−−++	2,6,10	17.19	±1.01
16,48,74	−+++−+−+−+−+−+−+−+−+−+−+	2,6,10	20.45	±0.90
17,53,84	+−−+−−−++++−−−−++++−++−−	3,7,11	22.23	±1.37
18,50,85	−+−−−+−+−−+−+−+−++−+−+−+	3,7,11	17.60	±1.23
19,49,82	+−+++−−++−−++−−++−−+−−++	3,7,11	19.30	±0.64
20,56,87	−++−++−+−+−+−−+−+−+−+−+−	3,7,11	18.20	±1.08
21,55,88	+−++−++−−++−−++−−++−−−++	3,7,11	19.75	±0.96
22,52,83	+−−++++−−−−++++−−−−+++−−	3,7,11	15.61	±0.45
23,54,86	−+−−+−+−++−+−+−+−−+−−+−+	3,7,11	17.73	±1.15
24,51,81	−++−−−+−+−+−++−+−+−++−+−	3,7,11	18.16	±1.15
25,62,90	++−++−−−−++++−−−−+++−−−−	4,8,12	21.27	±0.82
26,58,93	−−+−−+−−++−−+−++−−++−++−	4,8,12	15.68	±0.49
27,59,91	++++++++++++++++++++++++	4,8,12	25.24	±1.56
28,60,94	++−+−++++−−−−++++−−−−−−−	4,8,12	20.77	±1.14
29,64,95	−−−−−−++−+−−++−−+−+++−−+	4,8,12	15.84	±0.62
30,57,92	−−−−++−−+−++−−++−+−−+−−+	4,8,12	16.74	±0.45
31,61,96	++++−−−−−−−−−−−−−−−−++++	4,8,12	17.86	±1.03
32,63,89	−−+−+−++−−++−+−−++−−−++−	4,8,12	16.52	±0.38

The symbol “−” represents the low level of a factor and “+” represents the high level of a factor for each of the 24 factors tested.

**Table 3 life-06-00035-t003:** Multiple linear regression analysis of 1,3-propanediol (PDO) yield by using JMP Pro 12.1 software for each assay run.

Factor	Number of Parameters	Degrees of Freedom	Sum of Squares	F Ratio	Prob > F
GLYC	1	1	55.161176	67.9746	<0.0001
B_12_	1	1	92.022084	113.3979	<0.0001
pH	1	1	4.964051	6.1172	0.0164
NH_4_	1	1	67.318251	82.9556	<0.0001
NO_3_	1	1	0.040426	0.0498	0.8242
SO_4_	1	1	0.259376	0.3196	0.5741
PO_4_	1	1	0.587501	0.724	0.3985
NAK	1	1	1.226276	1.5111	0.2241
NAS	1	1	60.055884	74.0063	<0.0001
CYS	1	1	51.993984	64.0717	<0.0001
CL	1	1	0.681751	0.8401	0.3633
CA	1	1	2.145026	2.6433	0.1096
MG	1	1	1.318359	1.6246	0.2077
SI	1	1	0.100751	0.1242	0.7259
MN	1	1	0.008626	0.0106	0.9183
ZN	1	1	0.106001	0.1306	0.7191
FE	1	1	46.078959	56.7826	<0.0001
CO	1	1	86.165651	106.1811	<0.0001
BO_3_	1	1	0.958001	1.1805	0.2819
NI	1	1	0.221376	0.2728	0.6035
M_O_O_4_	1	1	0.111384	0.1373	0.7124
WO_4_	1	1	0.090651	0.1117	0.7395
CU	1	1	0.197109	0.2429	0.624
AL	1	1	0.094376	0.1163	0.7344
Block	11	11	7.465253	0.8363	0.6052
GLYC * pH	1	1	0.469001	0.5779	0.4503
B_12_ * pH	1	1	0.198926	0.2451	0.6225
pH * M_O_O_4_	1	1	0.098176	0.121	0.7293
NH_4_ * M_O_O_4_	1	1	1.547876	1.9074	0.1727

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

*acetobutylicum* alcohol dehydrogenase gene in *Escherichia*
*coli*. Appl. Environ. Microbiol..

