# Peer review of "The Use of a Fractional Factorial Design to Determine the Factors That Impact 1,3-Propanediol Production from Glycerol by Halanaerobium hydrogeniformans"

_life, 2016, doi:10.3390/life6030035_

Reviewer 1 Report

The work is of good quality but I suggest to the authors to insist more  on microorganisms able of carrying out this reaction in Introduction part.

Author Response

Thank you for your suggestions.  We have added information on Citrobacter freundii and Halanaerobium saccharolyticum subsp.  saccharolyticum in our introduction based on your suggestion.

Reviewer 2 Report

This study examines the problem of purifying/processing crude glycerol prior to biofuel applications. H. hydrogeniformans has a broad pH and salt tolerance, and the ability to convert crude glycerol to PDO. The authors suggest that this species, due to its PDO biochemical pathway and flexibility in growth conditions, could be a candidate for production of biofuels. They examined conditions and reagents that impacted the PDO pathway and report here on eight factors that may be useful in affecting PDO production, underscoring the potential of H. hydrogeniformans and this biotech application. The statistical analysis is robust.

The design, incorporating an analysis of the genetic potential based on genome data, is very smart. It is clear in the paper that there is logic behind the choice of each tested factor. Using a simple growth assay that illuminates PDO production, the authors tested 24 factors. Table 1 is exceedingly helpful in organizing all of these and in helping the reader understand the tested range (High and Low) of all of these.

Suggested Edits:

·      Table 1: Where applicable, the authors should include references for each of the tested reagents (especially concentrations) and their use in bacterial growth. Otherwise the High and Low values seem arbitrary. This would help the reader follow the logic behind the choice of concentrations (High and Low).

·      Table 2: Include units in Table for PDO yield (perhaps in heading for that column). Units are defined in the text, but it would be nice to have in the table.

·      Figure 1 did not show up in my pdf, may be a technical problem to correct as I can only see the heading on the figure legend. But I think the PDO pathway is important to have in this paper.

·      Small Edits by line number:

18. Clarify “without the need for salt dilution or adjusting pH,” it’s unclear if you are talking about for growth, production or purification in this phrase. After the reading the paper, it is clear to me, but the abstract should stand alone.

33. Insert commas: …resources, i.e. fossil fuels, are…

33-34. Rephrase or replace “has drawn interest from [the] scientific community to develop” seems grammatically clunky.

36. “amount” should be “amounts” or “a large amount”

47. Delete “subsequently” as indicates a timeline. Maybe just start sentence with “Effective methods for the utilization…”

59. Should “salt” be “sodium chloride?” or “total salts?”

64-70. Any references on genome information? Should Figure 1 be referred to here? (I cant see the figure, but it seems appropriate)

74. Insert “the” in “Therefore, THE primary objective…”

108. Heading “2.2. Experimental Design” should be more descriptive.

114. Heading “2.3. Statistical Design of Experiments” should be more descriptive.

161. Clarify/modify: “an experimental design was generated” and “this experimental design was created…” are repetitive and not very specific.

174.  Insert comma after “is accepted”

184. “This experimental design” should be more specific.

198. Any references on the genome information?

238-244. I think the “Conclusions” section for the manuscript is optional for this journal, and I would suggest that you don't need it here. Your discussion ends well, and this section really doesn't add anything.

Author Response

Thank you for your suggested edits!  They greatly improved the manuscript.

For your suggested edits:

Table 1.  We didn't include references in the table since it is already very busy.  However, we added a reference to the medium that we use (line 93) and provided the rationale for the high and low values used (lines 121-122).

Table 2.  The unit (percentage of initial glycerol converted to PDO) was added to Table 2.

Figure 1.  Initially, we were not going to include this figure and we removed it.  However, the title should have been deleted but must have slipped behind the text instead.  A fortunate mistake!  We have now included the figure with the PDO pathway.

Small Edits by Line Number:

18.  We added the words "when grown on this waste".

33.  The commas have been inserted as recommended.

33-34:  The sentence has been rephrased to "To address this problem, the scientific community is working to develop effective methods..."

36:  Changed to "amounts".

47:  The word "Subsequently" has been removed.

59:  We have replaced "salt" with "NaCl".

64-70:  The genome reference has been added as well as a reference to Figure 1.

74:  "the" has been inserted.

108 and 114:  Each of these headings have been made more descriptive.

161:  This section has been rewritten to be less repetitive and clearer.

174:  A comma has been inserted after "is accepted".

184:  This sentence has been rewritten to be more specific.

198:  A reference on the genome information has been inserted.

238-244:  Our conclusions section has been eliminated.